# Time to publish? Turnaround times, acceptance rates, and impact factors of journals in fisheries science

**Brendan J. Runde**[ID]¤*

Department of Applied Ecology, North Carolina State University, Morehead City, North Carolina, United States of America

¤ Current address: Southeast Fisheries Science Center, National Marine Fisheries Service, Beaufort, North Carolina, United States of America
* Brendan.Runde@noaa.gov

## Abstract

Selecting a target journal is a universal decision faced by authors of scientific papers. Components of the decision, including expected turnaround time, journal acceptance rate, and journal impact factor, vary in terms of accessibility. In this study, I collated recent turnaround times and impact factors for 82 journals that publish papers in the field of fisheries sciences. In addition, I gathered acceptance rates for the same journals when possible. Findings indicated clear among-journal differences in turnaround time, with median times-to-publication ranging from 79 to 323 days. There was no clear correlation between turnaround time and acceptance rate nor between turnaround time and impact factor; however, acceptance rate and impact factor were negatively correlated. I found no field-wide differences in turnaround time since the beginning of the COVID-19 pandemic, though some individual journals took significantly longer or significantly shorter to publish during the pandemic. Depending on their priorities, authors choosing a target journal should use the results of this study as guidance toward a more informed decision.

## Introduction

Settling on a target journal for a completed scientific manuscript can be a non-scientific process. Some critical elements of the decision are intangible, e.g., attempting to reach a certain target audience or how well the paper "fits" within the scope of the journal [1–3]. Others, such as turnaround time, acceptance rate, and journal impact, *can* be measured but (other than impact) these metrics are often challenging to locate, leading authors to make decisions without full information [3, 4].

Timeliness of publication has been reported as among the most important factors in the decision of target journal [4–8]. Prolonged peer review and/or production can be a major hindrance to authors [9]. Aarssen et al. [4] surveyed authors of ecological papers and found that 72.2% considered likelihood of a rapid decision a "very important" or "important" factor in choosing a journal. In some fields, research outcomes may be time-sensitive, so lengthy review

**Data Availability Statement:** All relevant data are within the manuscript and its Supporting information files.

**Funding:** The author(s) received no specific funding for this work.

**Competing interests:** The authors have declared that no competing interests exist.

can render results obsolete even before publication [10]. Desires and expectations for turnaround time are often not met: Mulligan et al. [11] found that 43% of survey respondents rated "time-to-first-decision" of their most recent article as "slow" or "very slow." Allen et al. [12] found that authors in the life sciences expect peer review to take less than 30 days (although this may be unrealistic). Moreover, Nguyen et al. [7] conducted a survey of authors in conservation biology in which the vast majority (86%) of respondents reported that their perceived optimal duration for peer review was eight weeks or under, though their experienced peer review time was on average 14.4 weeks. Over half of the respondents in Nguyen et al. [7] believed that lengthy peer-reviews can have a detrimental impact on their career, including individuals who reported that the lack of timely publication obstructed their acceptance into educational institutions and caused delays to degree conferral.

Despite the obvious and documented importance of journal turnaround time, published per-journal values are almost non-existent (BR, personal observation). Some journals do publicize "time-to-first-decision" on their (or their publisher's) webpages (e.g., *ICES Journal of Marine Science*), but summary statistics of times to acceptance and publication remain generally unavailable to the public. Lewallen and Crane [13] recognized the importance of turnaround time and recommended authors contact potential target journals and request information directly. However, this approach is time-consuming and unlikely to result in universal acquiescence from potential target journals. Moreover, because the duration of the review process is unpredictable, journals are more likely to give an average or a range—as an indicator—rather than guarantee a specific turnaround time (H. Browman, Ed. in Chief, *ICES J. Mar. Sci.*, personal communication).

In many biological journals, individual papers contain metadata that can be used to generate turnaround times. Specifically, a majority of journals in the sciences report "Date Received," "Date Accepted," and at least one of "Date Published," "Date Available," or similar on the webpage or in the downloadable PDF of each paper (BR, personal observation). Aggregating these dates on a per-journal basis allows for the calculation of turnaround time statistics, which would be extremely valuable to authors seeking to identify an ideal target journal.

In this study, I present summary data on turnaround times for over 80 journals that regularly publish papers in fisheries science and the surrounding disciplines. I restrict my analyses to this field out of personal interest and because cross-discipline comparisons may not be apt. Moreover, my goal in this study is to provide field-specific information, and data on journals in other disciplines was beyond that scope. In addition, I provide per-journal information on impact factor and acceptance rate (where available) which are also key factors in deciding on a target journal [4]. The information presented herein is intended to be used in concert with other factors, including authors' notions of their paper's "fit," to refine the process of selecting a target journal.

## Methods

### Literature review and journal selection

I began by developing a list of journals that regularly publish papers in fisheries science. On 20 March 2021, I searched the Web of Science Core Collection (Clarivate Analytics; v.5.35) for published articles with "fisheries or fishermen or fishes or fish or fishing" as the topic. These terms were used by Branch and Linnell [14] for a similar purpose. I refined this search by selecting only "Articles" and "Proceedings Papers" thereby excluding reviews, meeting abstracts, brief communications, et cetera. Finally, I truncated the search to include only documents that were published during 2010–2020. This search resulted in 242,280 published works. Using Web of Science's "Analyze Results" tool, I compiled a list of source titles (i.e.,

journals) that have published >400 papers meeting the specifics of my query. This threshold was used because it emerged as a natural break in the list of journals. A total of 85 journals met these requirements. I removed from this list journals that publish strictly in the field of food sciences (e.g., *Food Chemistry*) as well as hyper-regional journals that may not be of broad interest to authors in the field (though their exclusion is not indicative of their quality). Finally, I added several journals *ad hoc* that had not met the 400-paper minimum. These additions were included either because of my personal interest (e.g., *Marine and Coastal Fisheries* and *Global Change Biology*) or because of their relevance and value in among-journal comparisons (e.g., *Science* and *Nature*). After removals and additions, the list included 82 total journals.

**Turnaround time.** In the spring of 2021, I accessed webpages of each of the 82 journals selected for inclusion. For each journal, I located publication history information (i.e., dates received, accepted, and published) on the webpages or in the PDFs of individual papers. I tabulated these dates for each paper. Generally, I aspired to gather dates for all papers published from present day back to at least the beginning of 2018. It was my explicit goal to compare timeliness of publication only for original research papers. For all journals where possible, I excluded papers if they were not original research articles. Some journals publish a higher proportion of reviews, brief communications, errata, or editorials, all of which likely have a shorter turnaround time than original research. Most journals list the paper type on each document, allowing for easy exclusion of papers that were not original research.

I examined distributions of time-to-acceptance (calculated as *date accepted–date received*) and time-to-publication (calculated as *date published—date received*). For *date published*, I used the earliest date after acceptance, i.e., if "date published online" and "date published in an issue" were both provided, I used only "date published online." Some articles reported acceptance times that are inconsistent with the usual paradigm of peer review (for instance, progressing from *received* to *accepted* in 0 days). It is highly unlikely (perhaps impossible) that an unsolicited original research article could be accepted or published within 30 days of submission. I assumed that any implausibly short publication histories either were typographical errors, artifacts of that journal's methods for tracking papers, or the papers were simply not unsolicited original research articles. I therefore excluded from further analysis any papers with a time-to-acceptance or time-to-publication of fewer than 30 days; by-journal proportions of such papers ranged from zero to 0.06 (Table 1). Similarly, some papers reported publication times on the order of several years or more since receipt. While extreme delays in publication are certainly possible, I assumed that any paper with a time-to-publication of over 600 days was either a typographical error or a result of extenuating circumstances for which the journal staff and reviewers likely played no role. I therefore excluded papers with a time-to-acceptance or a time-to-publication of over 600 days from further analysis; by-journal proportions of such papers ranged from zero to 0.08 (Table 1). Paper-by-paper information on the duration from receipt until reviews are received is generally not available. However, this so-called "time-to-first-decision" is often available on journal websites. Where available, I obtained time-to-first-decision for each journal.

I generated summary data for each journal in this study in R [15]. Specifically, I examined median time-to-acceptance, median time-to-publication, median time between acceptance and publication, proportion of papers published in under six months, and proportion of papers published in over one year. For the latter two metrics, I selected six months and one year because, though arbitrary, these durations may be representative of many authors' notions of short versus long turnaround times. Medians were used because distributions of time-to-acceptance and time-to-publication were usually skewed right (see Results).

Some journals included in this study have an extremely broad scope. Specifically, *Nature*, *PeerJ*, *PLOS ONE*, *Proceedings of the National Academy of Sciences*, and *Science* publish papers

**Table 1. Publication histories for individual papers included in this study.**

| Journal | % < 30 d | % > 600 d | N | Start | Stop | T acc | T pub | % > 1 yr | % < 6 mo | COVID | COVID p | IF | Peer-reviewed | Overall accept | T first dec |
|---|---|---|---|---|---|---|---|---|---|---|---|---|---|---|---|
| Acta Ichthyologica et Piscatoria | 0.00 | 0.02 | 171 | 6/30/2017 | 12/7/2020 | 122 | 255 | 0.12 | 0.22 | -109 | < 0.05 | 0.72 | - | 0.24 | - |
| Aquaculture | 0.02 | 0.01 | 891 | 6/27/2019 | 6/24/2020 | 118 | 121 | 0.03 | 0.76 | -21 | < 0.05 | 3.23 | - | 0.32 | 34 |
| Aquaculture International | 0.00 | 0.00 | 320 | 9/15/2017 | 7/27/2020 | 198 | 214 | 0.06 | 0.38 | -25 | 0.36 | 1.66 | 0.27 | 0.16 | 48 |
| Aquaculture Nutrition | 0.00 | 0.00 | 416 | 6/22/2017 | 12/21/2020 | 148 | 202 | 0.06 | 0.39 | -16 | 0.16 | 2.41 | - | - | - |
| Aquaculture Research | 0.00 | 0.00 | 1061 | 7/8/2017 | 1/26/2021 | 128 | 156 | 0.04 | 0.61 | 1 | 0.54 | 1.69 | - | - | - |
| Aquatic Conservation: Marine and Freshwater Ecosystems | 0.00 | 0.04 | 441 | 8/10/2017 | 2/3/2021 | 228 | 323 | 0.27 | 0.02 | 23 | < 0.05 | 3.09 | - | - | - |
| Aquatic Toxicology | 0.02 | 0.00 | 784 | 10/28/2017 | 4/2/2021 | 93 | 98 | 0.00 | 0.88 | 50 | < 0.05 | 3.93 | - | 0.23 | 32 |
| Biological Conservation | 0.00 | 0.00 | 992 | 10/28/2017 | 3/23/2021 | 157 | 180 | 0.04 | 0.50 | 8 | 0.37 | 4.69 | 0.4 | 0.20 | 36 |
| BMC Genomics | 0.01 | 0.00 | 824 | 1/3/2020 | 2/28/2021 | 164 | 182 | 0.05 | 0.49 | -19 | 0.62 | 3.53 | - | - | 60 |
| Bulletin of Environmental Contamination and Toxicology | 0.02 | 0.00 | 762 | 11/16/2017 | 3/4/2021 | 120 | 130 | 0.01 | 0.77 | 4 | 0.67 | 1.74 | - | 0.18 | 37 |
| Bulletin of Marine Science | 0.00 | 0.02 | 113 | 8/9/2017 | 11/2/2020 | 194 | 221 | 0.07 | 0.27 | 9 | 0.26 | 1.40 | 0.73 | 0.47 | 45 |
| Canadian Journal of Fisheries and Aquatic Sciences | 0.00 | 0.00 | 251 | 4/8/2017 | 2/25/2021 | 156 | 174 | 0.02 | 0.52 | -11 | 0.72 | 2.47 | - | 0.27 | 41 |
| Chemosphere | 0.01 | 0.00 | 7036 | 8/15/2017 | 2/26/2021 | 89 | 93 | 0.00 | 0.91 | -3 | < 0.05 | 5.35 | - | 0.27 | 27 |
| Comparative Biochemistry and Physiology Part A | 0.06 | 0.00 | 347 | 10/17/2017 | 3/11/2021 | 89 | 96 | 0.01 | 0.91 | -7 | 0.22 | 2.24 | - | 0.42 | 19 |
| Comparative Biochemistry and Physiology Part B | 0.05 | 0.00 | 302 | 7/16/2017 | 3/20/2021 | 96 | 106 | 0.00 | 0.92 | 7 | 0.55 | 2.09 | - | - | 14 |
| Comparative Biochemistry and Physiology Part C | 0.05 | 0.00 | 425 | 8/18/2017 | 3/3/2021 | 77 | 85 | 0.00 | 0.99 | 9 | 0.18 | 2.70 | - | 0.25 | 13 |
| Conservation Biology | 0.00 | 0.01 | 261 | 6/9/2017 | 12/22/2020 | 177 | 216 | 0.13 | 0.36 | 17 | 0.65 | 6.09 | 0.33 | 0.15 | 55 |
| Copeia / Ichthyology and Herpetology | 0.00 | 0.08 | 73 | 9/1/2017 | 12/28/2020 | 173 | 270 | 0.15 | 0.15 | 38 | 0.17 | 1.02 | 0.93 | 0.74 | 51 |
| Deep-Sea Research Part I | 0.00 | 0.00 | 396 | 11/15/2017 | 3/12/2021 | 180 | 189 | 0.08 | 0.46 | 2 | 0.77 | 2.82 | - | - | 46 |
| Deep-Sea Research Part II | 0.02 | 0.03 | 243 | 6/4/2017 | 12/4/2020 | 269 | 274 | 0.18 | 0.21 | 55 | < 0.05 | 2.70 | - | | |
| Developmental and Comparative Immunology | 0.05 | 0.00 | 682 | 4/19/2017 | 4/3/2021 | 74 | 79 | 0.00 | 0.96 | 1 | 0.47 | 3.13 | - | 0.46 | 22 |
| Diseases of Aquatic Organisms | 0.00 | 0.01 | 355 | 3/22/2018 | 3/25/2021 | 145 | 218 | 0.07 | 0.29 | -18 | 0.16 | 1.71 | - | - | - |
| Ecological Applications | 0.00 | 0.00 | 501 | 6/27/2017 | 11/6/2020 | 166 | 213 | 0.07 | 0.34 | 2 | 0.63 | 4.43 | 0.50 | 0.23 | - |
| Ecological Indicators | 0.00 | 0.01 | 1648 | 6/1/2017 | 3/18/2021 | 167 | 184 | 0.06 | 0.48 | -7 | 0.07 | 4.80 | - | 0.30 | 43 |
| Ecological Modelling | 0.00 | 0.01 | 805 | 11/5/2017 | 4/8/2021 | 153 | 175 | 0.06 | 0.53 | -1 | 0.38 | 2.75 | 0.50 | 0.28 | 39 |
| Ecology | 0.00 | 0.00 | 423 | 10/30/2018 | 12/16/2020 | 166 | 210 | 0.05 | 0.38 | 8 | 0.06 | 3.99 | 0.50 | 0.20 | - |
| Ecology and Evolution | 0.00 | 0.00 | 3077 | 9/5/2017 | 3/3/2021 | 124 | 175 | 0.06 | 0.52 | 4 | 0.28 | 2.54 | - | - | - |
| Ecology of Freshwater Fish | 0.00 | 0.00 | 132 | 6/27/2018 | 2/10/2021 | 138 | 180 | 0.05 | 0.50 | -18 | 0.46 | 1.68 | - | - | - |

*(Continued)*

**Table 1.** (Continued)

| Journal | % < 30 d | % > 600 d | N | Start | Stop | T acc | T pub | % > 1 yr | % < 6 mo | COVID | COVID p | IF | Peer-reviewed | Overall accept | T first dec |
|---|---|---|---|---|---|---|---|---|---|---|---|---|---|---|---|
| Ecotoxicology and Environmental Safety | 0.00 | 0.00 | 4154 | 8/16/2017 | 4/11/2021 | 100 | 113 | 0.00 | 0.88 | 5 | < 0.05 | 4.71 | - | 0.26 | 27 |
| Environmental Biology of Fishes | 0.00 | 0.01 | 223 | 6/27/2018 | 2/23/2021 | 208 | 232 | 0.13 | 0.29 | -57 | < 0.05 | 1.29 | - | 0.29 | 60 |
| Environmental Monitoring and Assessment | 0.00 | 0.00 | 801 | 12/5/2017 | 3/22/2021 | 173 | 190 | 0.04 | 0.43 | -7 | 0.84 | 2.10 | - | 0.19 | 56 |
| Environmental Pollution | 0.00 | 0.00 | 4086 | 9/13/2017 | 4/7/2021 | 111 | 120 | 0.00 | 0.84 | 4 | < 0.05 | 5.95 | - | 0.24 | 25 |
| Environmental Science and Pollution Research | 0.00 | 0.00 | 2598 | 10/5/2017 | 2/12/2021 | 130 | 148 | 0.02 | 0.65 | -17 | < 0.05 | 7.27 | - | - | - |
| Environmental Science and Technology | 0.01 | 0.00 | 1874 | 9/14/2018 | 6/1/2020 | 101 | 103 | 0.00 | 0.93 | 6 | 0.14 | 3.00 | - | 0.31 | 58 |
| Environmental Toxicology and Chemistry | 0.02 | 0.00 | 766 | 8/2/2017 | 1/7/2021 | 114 | 122 | 0.01 | 0.81 | 6 | 0.49 | 3.41 | - | - | - |
| Estuarine, Coastal, and Shelf Science | 0.01 | 0.01 | 1270 | 3/8/2017 | 4/7/2021 | 187 | 196 | 0.09 | 0.42 | 2 | 0.42 | 2.76 | - | 0.40 | - |
| Fish and Fisheries | 0.00 | 0.00 | 209 | 6/26/2017 | 2/27/2021 | 143 | 189 | 0.02 | 0.45 | -33 | < 0.05 | 6.37 | 0.46 | 0.22 | - |
| Fish and Shellfish Immunology | 0.01 | 0.00 | 1970 | 7/27/2017 | 3/4/2021 | 92 | 96 | 0.00 | 0.93 | 4 | 0.21 | 3.37 | 0.72 | 0.46 | 36 |
| Fish Physiology and Biochemistry | 0.00 | 0.01 | 448 | 7/21/2017 | 11/27/2020 | 162 | 185 | 0.11 | 0.48 | -9 | 0.28 | 1.64 | 0.45 | 0.25 | 65 |
| Fisheries | 0.00 | 0.02 | 54 | 12/14/2017 | 3/4/2021 | 178 | 188 | 0.15 | 0.39 | 72 | 0.50 | 1.79 | 0.90 | 0.82 | 17 |
| Fisheries Management and Ecology | 0.00 | 0.02 | 128 | 6/13/2018 | 11/20/2020 | 188 | 232 | 0.15 | 0.29 | 3 | 0.87 | 0.96 | - | - | - |
| Fisheries Research | 0.00 | 0.01 | 822 | 4/14/2017 | 4/7/2021 | 162 | 178 | 0.05 | 0.50 | -5 | 0.66 | 2.32 | - | 0.30 | 47 |
| Fisheries Science | 0.00 | 0.00 | 262 | 11/4/2017 | 2/16/2021 | 125 | 155 | 0.03 | 0.63 | 11 | 0.51 | 1.01 | - | 0.28 | 45 |
| Fishery Bulletin | 0.00 | 0.00 | 101 | 9/14/2017 | 3/22/2021 | 225 | 241 | 0.03 | 0.16 | -45 | 0.12 | 0.91 | - | - | - |
| Freshwater Biology | 0.00 | 0.03 | 405 | 5/10/2018 | 3/6/2021 | 214 | 253 | 0.15 | 0.18 | -5 | 0.87 | 3.40 | 0.67 | 0.24 | - |
| Frontiers in Marine Science | 0.00 | 0.00 | 432 | 1/27/2020 | 11/19/2020 | 125 | 156 | 0.02 | 0.61 | -5 | 0.33 | 3.07 | - | 0.82 | - |
| General and Comparative Endocrinology | 0.01 | 0.00 | 700 | 4/17/2017 | 3/29/2021 | 130 | 133 | 0.01 | 0.73 | 13 | 0.09 | 2.43 | 0.79 | 0.44 | - |
| Global Change Biology | 0.05 | 0.00 | 1273 | 7/15/2017 | 3/9/2021 | 122 | 148 | 0.02 | 0.64 | -19 | 0.21 | 9.02 | 0.45 | 0.17 | 5 |
| Hydrobiologia | 0.00 | 0.00 | 754 | 5/27/2017 | 2/2/2021 | 172 | 183 | 0.06 | 0.48 | 10 | 0.64 | 2.28 | - | 0.32 | 53 |
| ICES Journal of Marine Science | 0.00 | 0.00 | 510 | 3/28/2017 | 1/8/2021 | 134 | 177 | 0.02 | 0.52 | 16 | < 0.05 | 3.26 | 0.53 | 0.31 | 45 |
| Journal of Applied Ichthyology | 0.00 | 0.04 | 235 | 9/27/2017 | 2/11/2021 | 144 | 175 | 0.06 | 0.52 | 19 | 0.27 | 0.91 | - | - | - |
| Journal of Experimental Biology | 0.01 | 0.00 | 526 | 10/30/2017 | 3/18/2021 | 116 | 129 | 0.01 | 0.76 | 22 | < 0.05 | 2.75 | 0.59 | 0.33 | - |
| Journal of Experimental Marine Biology and Ecology | 0.00 | 0.01 | 364 | 7/28/2017 | 4/10/2021 | 178 | 195 | 0.07 | 0.43 | 28 | < 0.05 | 2.35 | - | 0.26 | 41 |
| Journal of Fish and Wildlife Management | 0.00 | 0.05 | 116 | 12/1/2018 | 9/21/2020 | 183 | 193 | 0.08 | 0.39 | -19 | 0.18 | 1.15 | - | - | - |
| Journal of Fish Biology | 0.02 | 0.01 | 646 | 11/6/2017 | 12/1/2020 | 130 | 142 | 0.02 | 0.68 | -18 | < 0.05 | 2.04 | - | - | - |

(Continued)

**Table 1.** (Continued)

| Journal | % < 30 d | % > 600 d | N | Start | Stop | T acc | T pub | % > 1 yr | % < 6 mo | COVID | COVID p | IF | Peer-reviewed | Overall accept | T first dec |
|---|---|---|---|---|---|---|---|---|---|---|---|---|---|---|---|
| Journal of Fish Diseases | 0.00 | 0.00 | 404 | 7/5/2017 | 2/12/2021 | 64 | 112 | 0.00 | 0.93 | -5 | 0.71 | 1.90 | - | - | - |
| Journal of Great Lakes Research | 0.00 | 0.00 | 399 | 8/18/2017 | 3/10/2021 | 179 | 214 | 0.07 | 0.35 | -43 | < 0.05 | 2.28 | - | 0.50 | - |
| Journal of the World Aquaculture Society | 0.00 | 0.03 | 148 | 10/11/2017 | 1/26/2021 | 184 | 232 | 0.14 | 0.30 | 43 | < 0.05 | 1.57 | 0.33 | 0.19 | - |
| Limnology and Oceanography | 0.00 | 0.01 | 456 | 1/15/2019 | 3/15/2021 | 219 | 262 | 0.15 | 0.15 | 40 | < 0.05 | 4.35 | 0.74 | 0.39 | - |
| Marine and Coastal Fisheries | 0.00 | 0.01 | 104 | 7/5/2017 | 2/27/2021 | 176 | 244 | 0.16 | 0.19 | 27 | < 0.05 | 1.52 | 0.80 | 0.72 | 66 |
| Marine and Freshwater Research | 0.00 | 0.01 | 449 | 8/21/2017 | 1/20/2021 | 144 | 223 | 0.11 | 0.25 | -65 | < 0.05 | 1.86 | - | - | - |
| Marine Biology | 0.00 | 0.00 | 520 | 4/9/2017 | 3/10/2021 | 168 | 188 | 0.04 | 0.47 | -12 | 0.32 | 2.17 | - | 0.29 | 40 |
| Marine Ecology Progress Series | 0.00 | 0.01 | 1061 | 2/23/2018 | 3/18/2021 | 183 | 252 | 0.13 | 0.18 | -5 | 0.24 | 2.38 | - | 0.50 | - |
| Marine Environmental Research | 0.00 | 0.00 | 686 | 10/31/2017 | 3/6/2021 | 109 | 114 | 0.00 | 0.85 | 11 | < 0.05 | 3.42 | - | 0.30 | 28 |
| Marine Policy | 0.04 | 0.01 | 1115 | 5/24/2017 | 4/11/2021 | 170 | 189 | 0.08 | 0.46 | -38 | < 0.05 | 3.04 | - | 0.45 | 69 |
| Marine Pollution Bulletin | 0.01 | 0.00 | 1904 | 6/24/2017 | 3/20/2021 | 117 | 133 | 0.02 | 0.72 | 1 | 0.91 | 3.75 | - | 0.42 | 48 |
| Mitochondrial DNA Part A | 0.00 | 0.00 | 223 | 3/30/2017 | 2/25/2021 | 85 | 112 | 0.00 | 0.84 | -28 | 0.08 | 0.55 | - | - | 12 |
| Molecular Ecology | 0.00 | 0.04 | 889 | 7/26/2017 | 2/5/2021 | 174 | 195 | 0.14 | 0.46 | 26 | < 0.05 | 5.58 | 0.64 | 0.22 | - |
| Nature | 0.00 | 0.07 | 1485 | 2/11/2019 | 3/24/2021 | 224 | 281 | 0.28 | 0.18 | 4 | 0.82 | 24.36 | - | 0.08 | 8 |
| Neotropical Ichthyology | 0.00 | 0.05 | 142 | 8/17/2017 | 1/29/2021 | 182 | 246 | 0.17 | 0.27 | -38 | 0.58 | 1.37 | 1.00 | 0.35 | - |
| North American Journal of Fisheries Management | 0.00 | 0.00 | 218 | 12/11/2017 | 2/26/2021 | 156 | 184 | 0.05 | 0.48 | 25 | 0.10 | 1.08 | 0.88 | 0.84 | 53 |
| Ocean and Coastal Management | 0.01 | 0.01 | 893 | 6/16/2017 | 4/8/2021 | 179 | 196 | 0.07 | 0.40 | 24 | < 0.05 | 2.83 | - | 0.32 | - |
| Parasitology Research | 0.00 | 0.00 | 1010 | 11/8/2017 | 2/26/2021 | 127 | 146 | 0.04 | 0.67 | -18 | < 0.05 | 2.26 | - | 0.32 | 34 |
| PeerJ | 0.00 | 0.01 | 1847 | 9/20/2018 | 3/8/2021 | 126 | 167 | 0.03 | 0.57 | 9 | < 0.05 | 2.34 | - | 0.42 | 30 |
| PLOS ONE | 0.00 | 0.01 | 1895 | 12/20/2018 | 2/22/2021 | 142 | 168 | 0.05 | 0.55 | -5 | 0.50 | 2.87 | 0.77 | 0.47 | 43 |
| Proceedings of the National Academy of Sciences | 0.00 | 0.00 | 1518 | 11/13/2018 | 3/2/2021 | 127 | 176 | 0.03 | 0.52 | 20 | < 0.05 | 9.35 | 0.36 | 0.14 | 21 |
| Proceedings of the Royal Society B | 0.00 | 0.00 | 1569 | 10/11/2017 | 3/31/2021 | 70 | 96 | 0.00 | 0.97 | 3 | 0.10 | 4.24 | - | 0.25 | 18 |
| River Research and Applications | 0.00 | 0.03 | 355 | 10/6/2017 | 1/24/2021 | 173 | 217 | 0.12 | 0.33 | 12 | 0.88 | 2.07 | - | - | - |
| Science | 0.00 | 0.01 | 1351 | 2/22/2019 | 3/5/2021 | 167 | 210 | 0.14 | 0.37 | 8 | 0.62 | 20.57 | 0.20 | 0.07 | - |
| Science of the Total Environment | 0.02 | 0.00 | 12538 | 4/12/2017 | 2/1/2021 | 76 | 85 | 0.00 | 0.96 | 2 | 0.30 | 5.90 | - | 0.25 | 16 |
| Scientific Reports | 0.00 | 0.01 | 974 | 7/25/2018 | 2/19/2021 | 165 | 192 | 0.06 | 0.44 | -5 | < 0.05 | 4.12 | - | 0.45 | 24 |

(*Continued*)

**Table 1.** (*Continued*)

| Journal | % < 30 d | % > 600 d | N | Start | Stop | T acc | T pub | % > 1 yr | % < 6 mo | COVID | COVID p | IF | Peer-reviewed | Overall accept | T first dec |
|---|---|---|---|---|---|---|---|---|---|---|---|---|---|---|---|
| Transactions of the American Fisheries Society | 0.00 | 0.00 | 185 | 12/10/ 2017 | 1/12/ 2021 | 121 | 148 | 0.02 | 0.62 | -2 | 0.53 | 1.46 | 0.88 | 0.40 | 49 |

Information and summary values pertaining to each of 82 journals that publish papers in fisheries and related topics. *% < 30 d* and *% > 600 d* refer to the proportion of papers with a publication time of less than 30 days or greater than 600 days; these papers were excluded from the analysis. *N* refers to the number of papers examined in this study and does not include those excluded for extremely short or extremely long turnaround times. *Start* and *Stop* refer to the range of publication dates for the papers examined in month / day / year format. *T acc* and *T pub* are median times (d) from submission to acceptance and from submission to publication, respectively. *% > 1 yr* is the proportion of papers examined that took more than 365 days from submission to publication. *% < 6 mo* is the proportion of papers that took less than 180 days from submission to publication. *COVID* is the difference in median times (d) to publication between the periods 01 March 2019–29 February 2020 and 01 March 2020–28 February 2021; *COVID p* is the p-value for a Wilcoxon rank sum test for a difference in publication times between these two time periods. *IF* is 2018 Journal Impact Factor. *Peer-reviewed* is the proportion of papers that are sent for peer review (i.e., 1 minus the rate of desk rejections). *Overall accept* is the proportion of all submission that are eventually accepted. *T first dec* is the mean or median (depending on what was reported) time (d) to first decision. Hyphens are included where information was not available.

on topics reaching far beyond fisheries or ecology. I hypothesized that turnaround times of fisheries papers published in these journals may be dissimilar to turnaround times for these journals overall since internal editorial structure at the journals may differ among disciplines. I queried Web of Science for "fisheries or fishermen or fishes or fish or fishing" for each of these five journals individually, obtained turnaround times for the resulting papers, and compared median times to publication for fisheries papers and for all papers in each journal.

## COVID-19 pandemic effects

During the COVID-19 pandemic, some journals offered leniency to authors and reviewers when setting deadlines to account for the increased probability of extenuating personal or professional circumstances (B. Runde, personal observation). Because of this phenomenon, I hypothesized that turnaround times for each journal may be different prior to and after the start of the COVID-19 pandemic. Hobday et al. [16] showed that for seven leading journals in marine science, times in review were shorter in February–June 2020 as compared to the previous year. For each journal in my study, I compared times-to-publication of all papers published during the year prior to the pandemic (1 March 2019–29 February 2020) and the year following the beginning of the pandemic (1 March 2020–28 February 2021). As above, papers were excluded from this analysis if their time-to-publication was extremely short (< 30 days) or extremely long (> 600 days). I conducted two-sample Wilcoxon tests to examine for differences in publication times between these two periods. Significance was evaluated at the $\alpha$ = 0.05 level. Analyses were performed in R [15].

## Impact factors

The most widely used metric of impact, impact factor, is considered flawed by some scientists due to the disproportionate influence of review articles and its propensity for manipulation [17–19]. Nonetheless, impact factor is still listed on many journal webpages and is relied on by many authors [20–22]. I obtained impact factor for 2018 (the most recent year for which it was available for all journals) from https://www.resurchify.com/impact-factor.php. Impact factor is calculated as the number of citations received in a given year by all articles published in that journal during the previous two years, divided by the number of articles published in that journal during the previous two years.

## Acceptance rates

I searched the web for reliable (i.e., not anecdotal) information on per-journal acceptance rates, which was generally limited. Most journals reject a percentage of submissions at the editorial stage prior to peer review (so-called "desk rejections") due to a lack of fit within the journal's scope, deficiencies in writing quality, and/or insignificant scientific merit [23]. Of course, rejections after peer review also occur, and overall rejection rates are increasingly made available on journals' or publishers' websites or in compendium papers [e.g., 20]. Unfortunately, rates of desk rejections are still rarely available online [23]. However, many journals' overall acceptance rates are reported either on their own page or on the publisher's website. For instance, Elsevier and Springer both offer acceptance rates for some (but not all) of their journals on their *JournalFinder* (https://journalfinder.elsevier.com/) and *Journal suggester* (https://journalsuggester.springer.com/) respectively. I extracted reported acceptance rates wherever available and tabulated them per journal. In addition, I sent email correspondence to Editors-in-Chief and/or publishers of each of the journals included in this study asking for their journal's desk rejection rate and overall acceptance rate. When information was provided, it was tabulated on a per-journal basis. In some cases, acceptance rates provided via email were not equal to the rate provided on the journal's webpage. In these cases, the value provided by the editor or publisher was used, as it is likely more recent and thus more valid. Such chases did not differ in these figures by more than 10%. It is possible that there are discrepancies in the calculation of acceptance rates, e.g., resubmissions may be tabulated differently among journals. I made no attempt to account for these potential differences in the present study.

## Data analysis

I examined summary data for each journal and calculated correlations between median time-to-publication, difference in median publication time during COVID-19 as compared to the prior year, impact factor, and acceptance rate (where available). I plotted correlations using the R package 'corrplot' [24]. In addition, I plotted relationships between median time-to-publication and impact factor.

## Results

From the 82 journals in this study, I extracted publication information for 83,797 individual papers. Median times to acceptance ranged from 64 to 269 days and median times-to-publication ranged from 79 to 323 days (Fig 1). Turnaround times did not differ substantially for fisheries papers in any of the five broad-scope journals in this study (Fig 2); therefore, for the other analyses in this study data from these journals were not restricted to fish-only papers. The ranges of times-to-publication for each journal were generally broad (Fig 3); the middle 50% often spanned a range of 100 days or more. Distributions were typically skewed right. Virtually every journal in the study published one or more papers that took close to 600 days to publish (the maximum timespan retained in the analysis). Percentages of papers published in over one year ranged from 0 to 28%; percentages of papers published in under 6 months ranged from 2 to 99% (Table 1). Of 82 journals examined, 28 had significantly different (Wilcoxon p < 0.05) times-to-publication in the year following the start of the COVID-19 pandemic as compared to the previous year. Of these 28, 12 were significantly faster and 16 were significantly slower during the pandemic (Table 1).

I was able to obtain overall acceptance rate information for 60 journals in this study. Of these 60, I gathered desk rejection rates for 27 journals. For each of these 27, I calculated

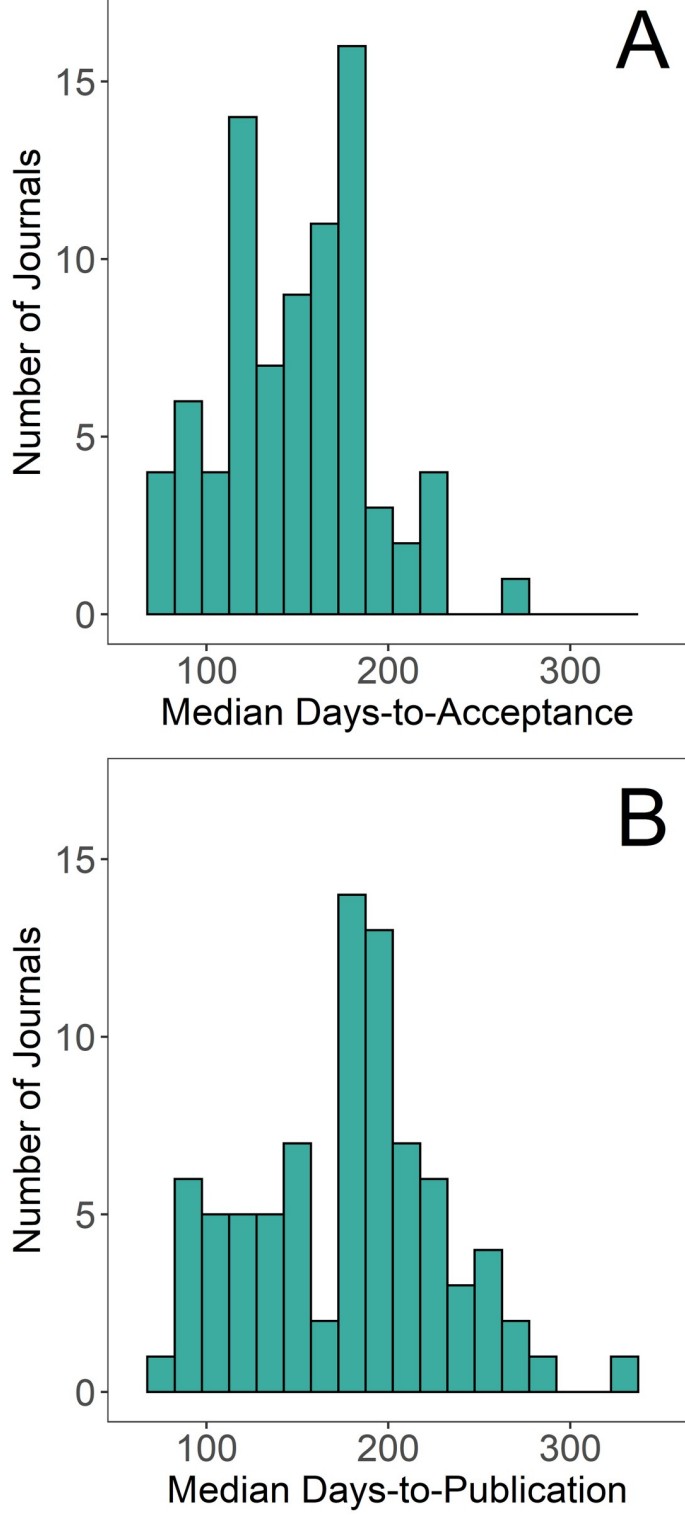

**Fig 1. Histograms of median days-to-acceptance (A) and median days-to-publication (B) for 82 journals that publish papers in fisheries and related topics.**

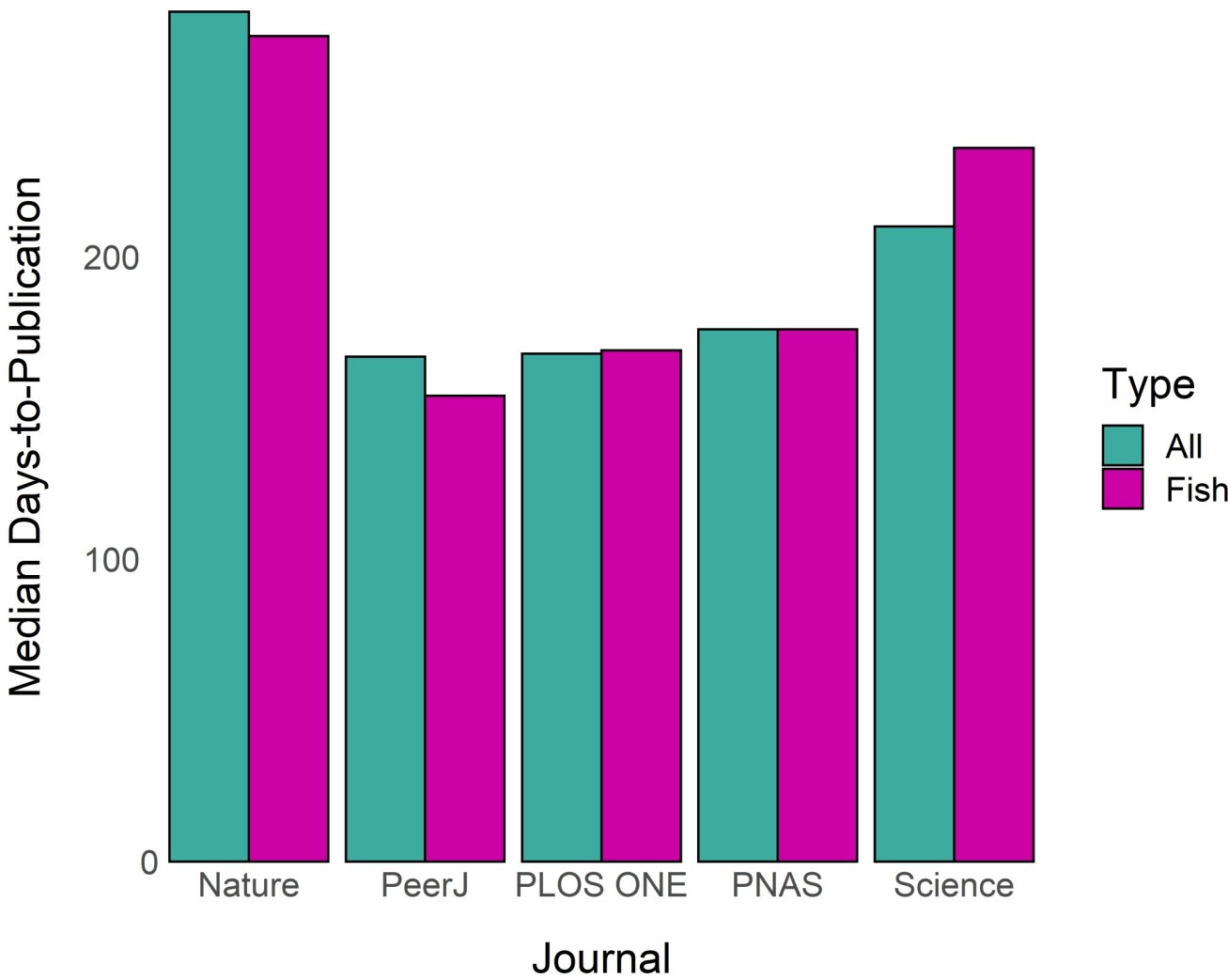

**Fig 2. Median days-to-publication for all papers (green) and papers related to fish or fisheries (pink) for five broad-scope journals included in this study.** PNAS is *Proceedings of the National Academy of Sciences.*

acceptance rates for papers that were peer-reviewed (i.e., not desk rejected). There was a weak positive correlation between this value and the proportion of articles that were peer-reviewed, implying that rates of the two types of rejections are not independent (Fig 4A). Higher impact journals tended to have higher desk rejection rates and lower percentages of acceptance given that peer review occurred. Of the 60 journals with overall acceptance rate information, I obtained time-to-first-decision for 48 journals; I plotted overall acceptance rate against these values (Fig 4B). There was no clear relationship between these variables; however, journals with higher impact tended to have lower acceptance rates and shorter times-to-first-decision.

There was no strong correlation between any pairwise combination of median time-to-publication, difference in median publication time during COVID-19 as compared to the prior year, impact factor, and acceptance rate (Fig 5). A moderate correlation (Pearson correlation = -0.43) was found between impact factor and overall acceptance rate, a phenomenon that has been documented previously [4]. The relationship between a journal's median time-to-publication and impact factor was broadly scattered (Fig 6).

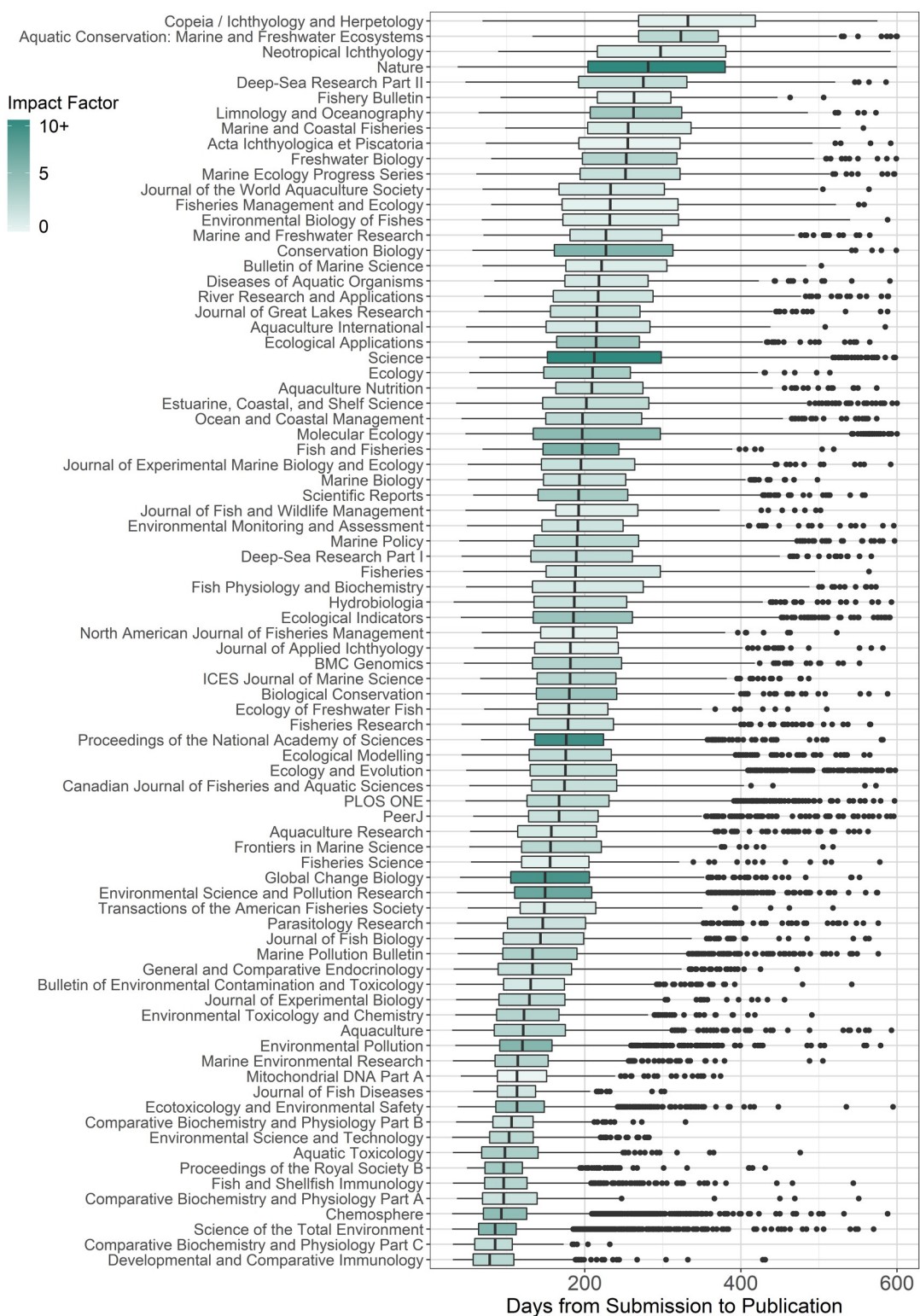

**Fig 3. Boxplots showing days from submission to publication for 82 journals that publish papers in fisheries and related topics organized in descending order of medians.** Central vertical lines represent medians, hinges represent the 25th and 75th percentiles, and lower and upper whiskers extend to either the lowest and highest values respectively or 1.5 * the inter-quartile range. Black dots represent papers that were outside 1.5 * the inter-quartile range. Boxes are shaded to correspond with 2018 Impact Factor, where darker green represents higher impact.

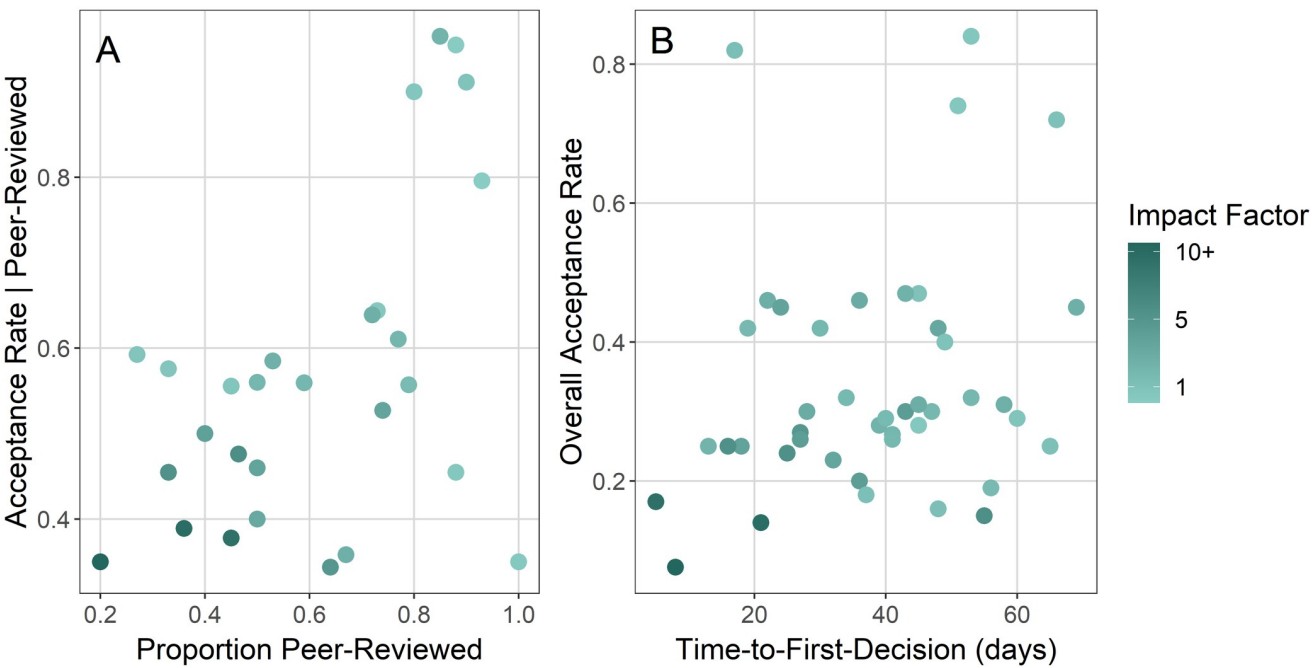

**Fig 4.** A) The proportion of submissions that are peer-reviewed (i.e., 1 minus the desk rejection rate) versus the acceptance rate of submissions given that they are peer-reviewed for 27 journals that publish in fisheries and related topics. B) Time-to-first-decision (d) versus overall acceptance rate for 48 journals that publish in fisheries and aquatic sciences. Points in both panels are shaded to reflect 2018 Impact Factor of each journal, where darker green means higher impact.

## Discussion

There are clearly intrinsic differences in turnaround time among journals that publish in fisheries science (Fig 3). The causes for these differences are varied, and some are artifacts of the journal's specific publishing paradigm. For instance, some journals publish uncorrected, non-typeset versions of accepted manuscripts very shortly after acceptance; for the purposes of this study, such papers were considered published even if they were not yet in their final form. I elected to consider any post-acceptance online version "published" because such versions can be shared and cited, thereby fulfilling the desires of many authors [7] and meeting one of the overall goals of science—disseminating research results. However, some journals do not publish any manuscript version other than the finalized document. Such journals have inherently longer turnaround times than those hosting unpolished versions online, and I made no attempt to specify or account for those differences in this study.

In addition to differences in which versions are published online first, differences in journal production formats can influence turnaround time. Some journals publish monthly, some publish quarterly, and some publish on a rolling basis (particularly those that are online only). Strictly periodical journals may choose to allow accepted papers to accumulate prior to publishing several in an issue all at once. Such journals, especially those with page limitations, may have a backlog of papers that are accepted but not yet published. I made no attempt to differentiate between journals based on these format differences, which certainly influence time-to-publication.

Similarly, some journals (or publishers) may enter revised manuscripts into their system as new submissions. This practice ostensibly artificially deflates turnaround times and may also artificially deflate acceptance rates. Unfortunately, to my knowledge no journals state publicly

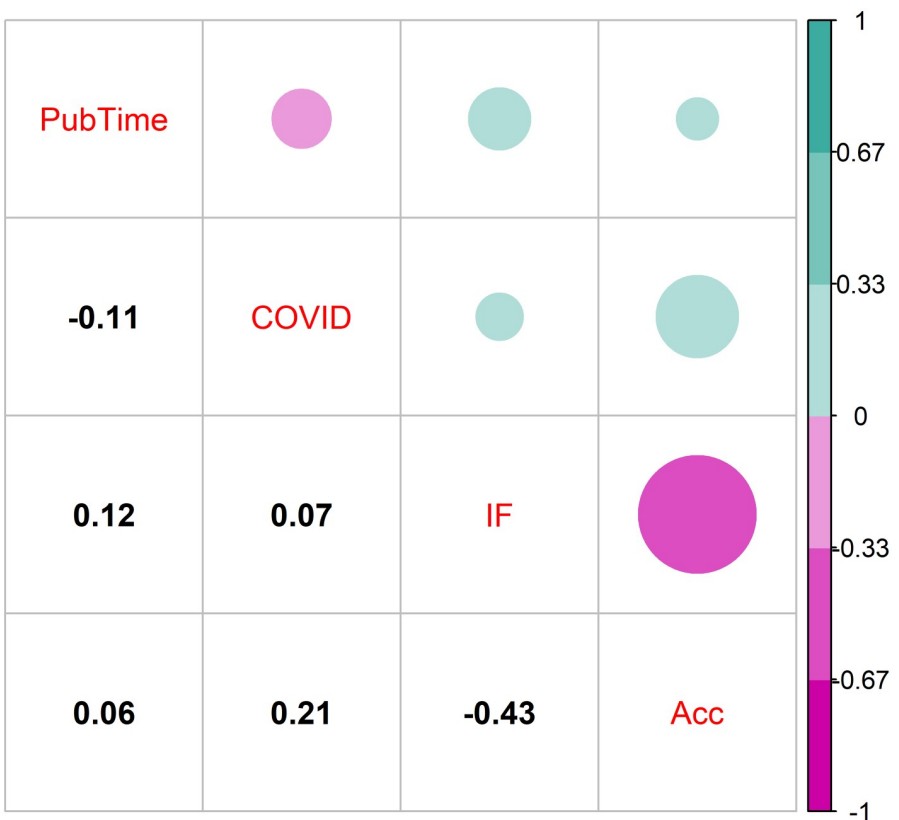

**Fig 5. Pearson correlations between time from submission to publication (*PubTime*; d), change in time from submission to publication since the start of the COVID-19 pandemic (*COVID*), 2018 Impact Factor (*IF*), and overall acceptance rate (*Acc*) for 61 journals that publish in fisheries and related topics (i.e., all journals in this study for which these four metrics were available).** Correlation bubbles are colored and shaded based on the calculated Pearson correlation coefficient, where negative correlations are pink, positive correlations are green, and darker shades and larger sizes represent stronger correlations.

whether this is their *modus operandi*, precluding the possibility of applying any correction factor or per-journal caveat herein.

Beyond these differences in production time that stem from journal structure, the time it takes to publish a paper can be divided into time the paper is with editorial staff, reviewers, and authors after review. Differences may exist in author revision time among journals; it is possible that reviews of manuscripts submitted to higher impact journals are more thorough and therefore require longer response times. However, I found no association between impact factor and turnaround time (Fig 6), so it may be that no such differences exist. Further, extenuating circumstances on the part of the author(s) of a paper may result in extremely lengthy revision times. There is no data available on per-journal rates of extension requests, but presumably it is low and approximately equivalent across journals. I removed from my dataset any papers that took longer than 600 days to publish. Still, I present median turnaround times in this study as a measure that is robust to outliers.

In contrast to time with the authors, it seems likely that among-journal differences in time with editorial staff and reviewers are responsible for a large portion of differences in overall turnaround time. Delays at the editorial and reviewer level may be inherent to each journal, and could be a result of editorial workload (i.e., number of submissions per editor), level of strictness of the editor-in-chief when communicating with the associate editors, or differences

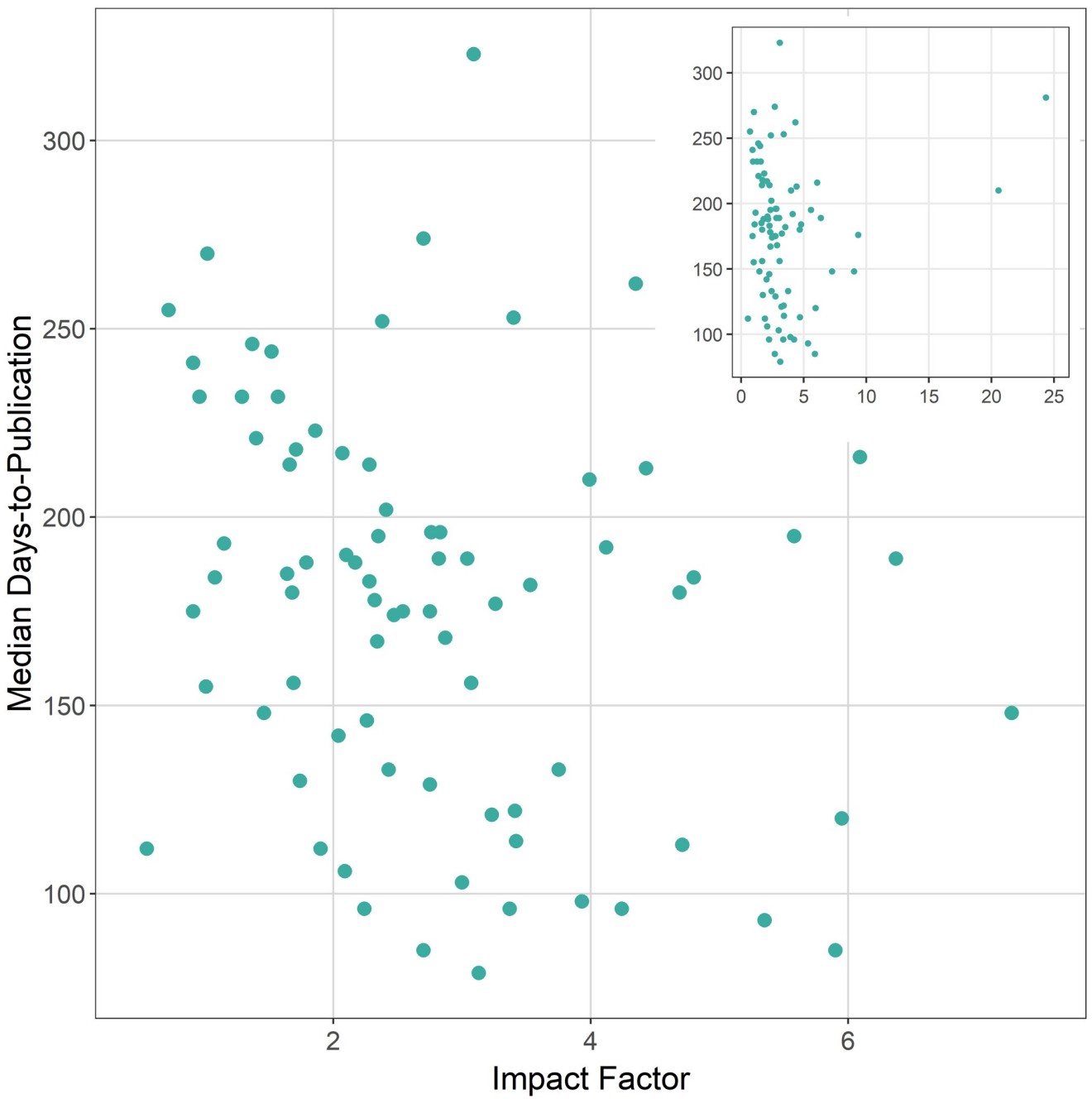

**Fig 6. Median time-to-publication (d) versus 2018 impact factor for 82 journals that publish in fisheries and related topics.** Inset panels shows a broader view to include *Science* and *Nature* which have high impact factors.

in persistence on the part of the editors when asking reviewers to be expeditious. In addition, some journals may have a more difficult time finding a suitable number of agreeable reviewers; this may be especially true for lower-impact journals although no association between IF and turnaround time was found. A majority of authors surveyed by Mulligan et al. [11] had declined to review at least one paper in the preceding 12 months, mainly due to the paper being outside the reviewer's area of expertise or the reviewer being too busy with work and/or

prior reviewing commitments. If among-journal differences do exist in acceptance rates of review requests, this could possibly alter turnaround times.

In this study, I treated impact factor as a proxy for the quality of individual journals. While impact factor is often still used in this way [22], its limitations are well-documented by authors across many disciplines [e.g., 25–27]. For instance, the calculation of how many "citable" documents a single journal has produced is often dubious, as this may or may not include errata, letters, and book reviews depending on the publisher [28]; misclassification can inflate or deflate a given journal's impact factor, and the rate of misclassification may depend on the individual journal's publishing paradigm [29]. Alternatives to impact factor, such as SCImago Journal Rank (SJR) and H-index, have been proposed and may in some cases be more valid metrics of journal prestige or quality [30, 31]. Comparison of these bibliometrics among journals in fisheries was beyond the scope of this paper, and I elected to use only impact factor given its ubiquity and despite its known disadvantages.

The COVID-19 pandemic had no discernable field-wide effect on turnaround time, and differences in turnaround time during the pandemic were not correlated with acceptance rate or impact factor (Fig 5). Hobday et al. [16] found minor changes in turnaround time during COVID-19 (through June 2020) for seven marine science journals; they reported only slight disruptions to scientific productivity in this field. Overall, my results corroborate those of Hobday et al. [16], although some journals took significantly longer or significantly shorter to

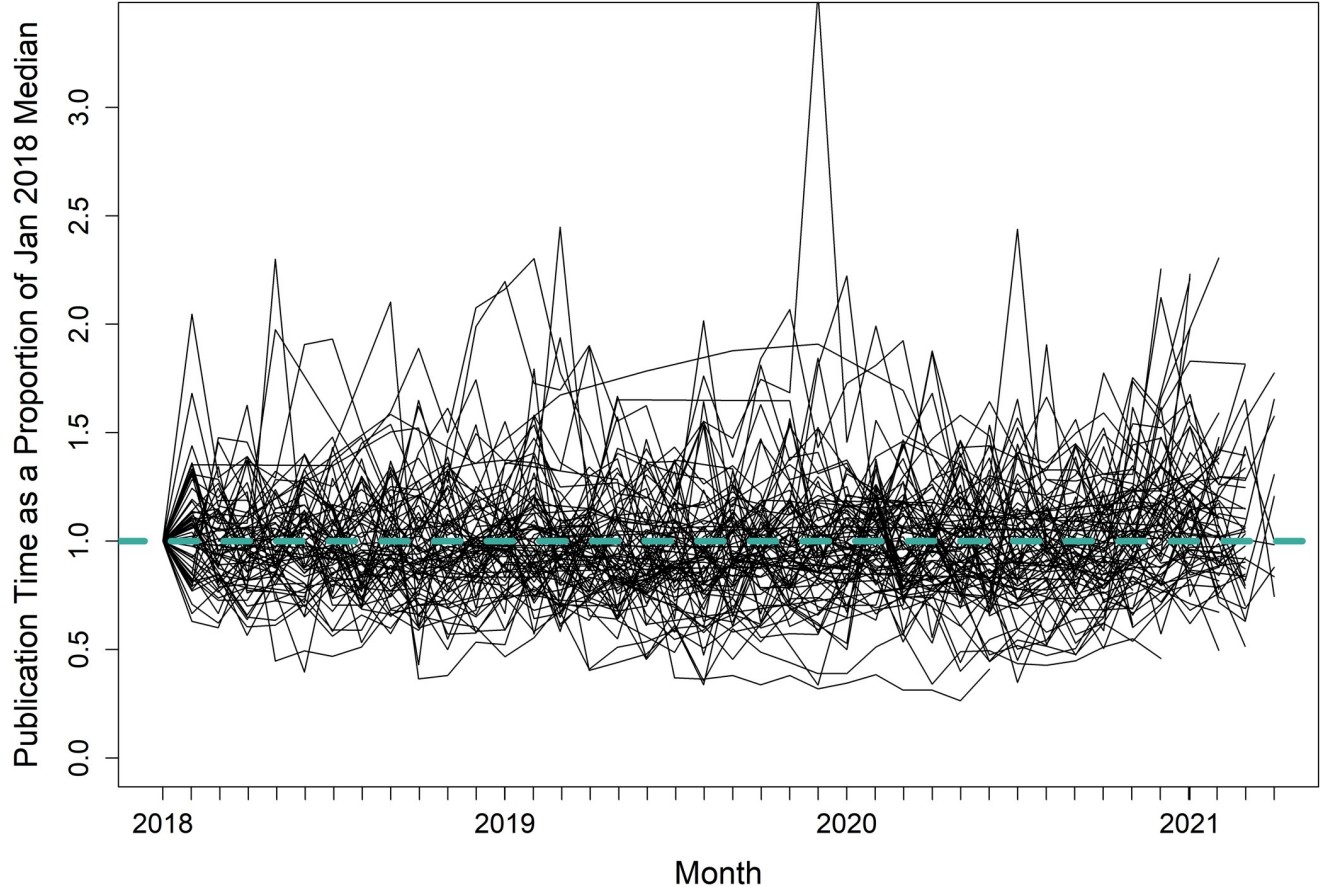

**Fig 7. Median monthly publication time (d) as a proportion of January 2018 median publication time for 82 journals that publish in fisheries and related topics.** The dashed horizontal line at 1.0 represents the baseline proportion.

publish during COVID-19. It is unclear whether these correlations were causal, as non-pandemic effects may have affected turnaround times at these individual journals.

The turnaround times, acceptance rates, and impact factors presented in this paper are snapshots and may change over time. The degree to which these metrics change is likely variable among journals. However, barring major changes in journal formats or editorial regimes, the data presented here are probably applicable for the next several years at least. Indeed, median monthly turnaround times for most journals in this study were approximately static for the period from January 2018 to April 2021 (Fig 7). Similarly, acceptance rates and impact factors [32] are generally strongly auto-correlated from one year to the next. I therefore suggest that the metrics presented here can be used by authors as a baseline, but if more than several years have transpired it may befit the reader to obtain updated information (particularly on impact factor and acceptance rate, which are generally more accessible than turnaround time). In addition, it is theoretically possible that this paper itself may alter turnaround times and/or acceptance rates for some journals. Enlightened readers may elect to change their submission habits in favor of certain journals that are more expeditious or that otherwise meet their priorities for a given paper. Authors without a preconceived notion of a specific target journal should still consider the paper's "fit" to be the most important factor in their decision [1]. I suggest that after assembling a shortlist based on fit, authors should use the results of this paper to select a journal that best aligns with their priorities.

## Supporting information

**S1 Data.**
(CSV)

## Acknowledgments

This manuscript benefited greatly from discussions with H. I. Browman, D. D. Aday, W. L. Smith, R. C. Chambers, N. M. Bacheler, K. W. Shertzer, S. R. Midway, S. M. Lombardo, and C. A. Harms. My thanks to K. W. Shertzer and H. I. Browman for reviewing early drafts of this paper. I am grateful to my advisor, J. A. Buckel, for allowing me the time to pursue this side project while I worked on my dissertation. Thanks to the numerous editors, publishers, and other journal staff who replied to my requests for journal information.

## Author Contributions

**Conceptualization:** Brendan J. Runde.

**Data curation:** Brendan J. Runde.

**Formal analysis:** Brendan J. Runde.

**Investigation:** Brendan J. Runde.

**Methodology:** Brendan J. Runde.

**Software:** Brendan J. Runde.

**Validation:** Brendan J. Runde.

**Visualization:** Brendan J. Runde.

**Writing – original draft:** Brendan J. Runde.

**Writing – review & editing:** Brendan J. Runde.

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
