## [Decision Letter · Decision Letter 0]

8 Sep 2021

PONE-D-21-22019

Time to publish? Turnaround times, acceptance rates, and impact factors of journals in fisheries science

PLOS ONE

Dear Dr. Runde,

Thank you for submitting your manuscript to PLOS ONE. After careful consideration, we feel that it has merit but does not fully meet PLOS ONE’s publication criteria as it currently stands. Therefore, we invite you to submit a revised version of the manuscript that addresses the points raised during the review process.

My remaining comment on the manuscript is that some discussion should be made on the limitations of using the journal impact factor as a proxy for individual articles' quality. Although this is briefly mentioned in the methods (L147-149), I suggest dedicating a paragraph in the discussion to the drawbacks of impact factors. A good location may be to add text between the 2nd and 3rd or 3rd and 4th paragraphs.

We look forward to receiving your revised manuscript.

Kind regards,

Charles William Martin

Academic Editor

PLOS ONE

Journal Requirements:

Reviewers' comments:

Reviewer's Responses to Questions

**Comments to the Author**

1. Is the manuscript technically sound, and do the data support the conclusions?

Reviewer #1: Yes

Reviewer #2: Yes

2. Has the statistical analysis been performed appropriately and rigorously? 

Reviewer #1: Yes

Reviewer #2: Yes

3. Have the authors made all data underlying the findings in their manuscript fully available?

Reviewer #1: Yes

Reviewer #2: Yes

4. Is the manuscript presented in an intelligible fashion and written in standard English?

Reviewer #1: No

Reviewer #2: Yes

5. Review Comments to the Author

Reviewer #1: In this study, the author compiled a number of publication metrics for a variety of fisheries or fisheries-adjacent journals and a select few other journals. Such metrics are relevant to the decision-making of authors of original research articles in journal selection. The study and compiled dataset therefore serve an immediately practical and useful service to anyone in fisheries and related fields trying to publish. The author also considered the relationships between the various collected metrics ant the COVID-19 pandemic. The compiled synthesis and analysis yielded interesting results, including some such as the limited COVID-19 impacts that defied the author’s presented hypothesis and my own expectations. The data collection process could serve as a guide to replicate the work on a wider scale or in a different field, leaving a template for others who may be interested in conducting such an effort. The study highlights the need for journals to report some of the metrics compiled themselves for the purposes of transparency.

The different decisions made by the author are clearly outlined and explained in the methods, so while I may have made slightly different decisions were I to conduct the work, I support the author’s decisions and subsequent analyses. The reasoning behind the statements in the discussion and introduction seemed largely reasonable and the figures and table all supported the results and objectives of the paper. The limitations of the study are clearly outlined, particularly in some of the speculations included in the discussion, which I appreciated. The inclusion of the raw table and Table 1 is also a wonderful example of data transparency.

I do have one general concern with the paper, which is that the author’s statements in the introduction and discussion are frequently not supported by citing literature. Many statements, such as almost all of the two paragraphs from lines 45-58, did not have citations to support various statements. It is possible the author made these statements based on their own experience working through their data collection, but if that were the case, I would like to see that more clearly stated or that some of the statements in the introduction should be moved to the discussion. This could also be alleviated by adding more citations to support various statements or at the very least by adding examples of journals that have the different policies or behaviors described by the author.

I have also identified more specific changes I would like to see made, outlined below. “L” means line numbers in the manuscript. I do not think that any of these issues deserve the article being rejected, just that the recommended revisions would improve the final published paper were it accepted.

L 35, 48, and elsewhere: The author is inconsistent about how they refer to authors from studies they are citing (e.g., Mulligan, Hall vs. Lewallen and Crane). I could not find PLoS ONE’s format guidelines for this situation, but I would prefer an “and” instead of a “comma.” Regardless, the authors should fix this throughout so it is at least consistent.

L 78: I appreciate the use of examples throughout this paragraph and wonder if more examples to some of the sentences in the introduction and discussion such as L 46 “Some journals…” to help contextualize the different assertions made and give readers a place to go to for examples if needed.

L 81: A comma should follow any use of e.g. or i.e. so this should be corrected throughout.

L 96-97, 113, 115-121, and elsewhere: The authors should remain consistent about the hyphens used in defining terms (e.g., time-to-acceptance, time-to-publication, time to first decision). The hyphens also disappear from some terms after they are defined as having them and this should also be kept consistent. I personally like the hyphened versions better but would accept either provided it was consistent throughout.

L 137: An “en” dash should be used for date ranges as is done on L 139, 140

L 147: If an acronym is going to be used for impact factor, it should be introduced and used after its first appearance in the main text when it’s mentioned in the introduction, not here in the methods. The acronym should then be used throughout, which it is not really outside of this paragraph. Also, impact factor is capitalized here when it is usually lower case in the manuscript, so this should also be kept consistent. I personally think impact factor could be lowercase and not acronymized throughout but would accept another option as long as it was consistent.

L 151-154: This definition is a little difficult to follow the way it is written out. Maybe it would be clearer if this were broken up into two sentences or if the order of some phrases were rearranged, but I’m not sure how much that would help.

L 169: I believe it should be written as “Editors-in-Chief.”

L 173-174: Change sentence to read “Such cases did not differ in these figures by more than 10%.” or alternatively “No such cases differed in these figures by more than 10%.”

L 203: While I believe I am correct in assuming that these 48 journals were part of the group of 60 journals with acceptance rates collected, it would be nice if that could be stated more clearly in the sentence, such as “Of the 60 journals with overall acceptance rate information, I obtained….”

L 204-205: Change to “…these variables; however, journals…”

L 219-221: This should be mentioned in the methods as well as here.

L 254: Change to “editor-in-chief”

L284-287: This paragraph is only 2 sentences long and I think could easily be combined with the paragraph above it.

Figures 1-6: Since colors were used in the figures, I would like to see them consistent so that one color is associated with a particular metric throughout all of the figures as opposed to green, then pink just being the default for all metrics. If the author chose to not use color, then using black then gray throughout would be an alternative option to multiple colors. While I was unable to confirm this because it is dependent on the exact shading, I also suspect the green and pink combination may be challenging colors to differentiate for colorblind readers. The author could avoid this potential issue by using colorblind friendly palettes, as are available and easily accessible, particularly in R if that’s how the figures were made.

Figure 1: Since this figure is made up of two separate graphs, not in a paneled style, I think the graphs should be broken up into Figure 1A and Figure 1B.

Figure 3: The legend reads “times to publication” but the axis in the figure reads “days from submission to publication” and I think these should be made consistent.

Figure 4: While “peer review” as a noun does not need hyphens, “peer-reviewed” is an adjective and therefore does need hyphens so this should be corrected here and throughout if there are other similar instances.

Figure 7: I think this is the only instance where median is placed in parentheses as opposed to in front of the particular metric, so I think this should be changed to read “Median monthly publication time….”

Figure 4: Here, the axis reads “Time to First Decision (days),” but the other axes have “days” outside parentheses, so I think that should be kept consistent.

Figure 7: The y axis should be all capitalized to be consistent with the other axes.

Figures 1-7 and Table 1: I don’t think the author should use the term “fisheries and aquatic sciences” to describe the journals included. Aquatic sciences could include a much wider variety of topics, from aquatic plant physiology to oceanography, that would not have been included in this study. Perhaps “fisheries and related topics” may be more accurate or something similar. This phrase is also only used in the legends, not anywhere else in the paper, as another reason why it should be altered.

Reviewer #2: Summary

The manuscript presents a concise analysis of turnaround times for journals in fisheries science. The paper is well written and logically structured, and frankly enjoyable to read.

I have no outright major concerns. Most of my comments and questions regard formatting conventions and clarity, particularly in the presentation of the methods and results.

Title is clear and appropriate.

Abstract is well written.

Line-specific comments:

31: Aarssen et al. Unless “Aarssen, Trengenza (4)” is acceptable formatting for PLOS One, this looks like only two authors are on this manuscript (or a typo).

35: Mulligan et al. Check (and correct) this formatting throughout the manuscript.

38: May be worth mentioning that Nguyen et al. (7) surveyed researchers in conservation biology. In fact, this may be useful context for the other statistics cited here (like you mentioned for Aarssen and Tregenza (4)).

61-62: Why wouldn’t cross-discipline comparisons “be apt”? Maybe more specifically, this is beyond the scope of your objective, which is to provide a sort of fact sheet of turnaround times for authors in fisheries sciences to consider when deciding on a journal?

74: Why only 2010-2020? Electronic versions were available in the 2000s as well.

76: Again, why the >400 threshold? It may be clearer to bullet point or tabulate all the specific criteria used in your query to create your dataset.

90: I thought papers since 2010 were considered, but this says papers published since 2018? What is the actual criterion (and sample size) for analysis?

132: Was this demonstrated or directly observed (that “many journals offered leniency…”)? Perhaps this is better posited as a hypothesis or suspected phenomenon based on your own observations.

137: Times in review were shorter during the beginning of the pandemic than before? This seems counterintuitive to the line of reasoning implied in the previous sentences (journal leniency in deadlines leading to extended turnaround times). Please clarify or elaborate on your hypothesis.

147-149: The topic sentence is somewhat misleading for the paragraph (e.g. IF is flawed, but I use it anyway). Suggest introducing IF as most widely used and easily accessed metric, and moving the current topic sentence to later in the paragraph as a caveat.

171-172: This makes me wonder if there is a universal standard/definition for how each journal calculates acceptance rates. Is it the number of papers accepted out of the total number of submissions? Does it account for resubmissions that were invited after an initial rejection (or more common practice nowadays)? Suggest stating this caveat to these statistics.

189-190: Could you be more precise than “often”? For example, how many journals had the middle 50% span at least 100 days?

200-201 & Fig. 4: “Negative”? First, Figure 4A shows acceptance rate VS proportion peer review. Second, if looks like there would be a positive correlation between these two values if any relationship. I see the color scale reflects Impact Factor, which looks to have a negative correlation with acceptance rate. My dissection digression aside, I suggest directly plotting the two quantities (acceptance and desk rejection rate, or impact factor with rejection and acceptance rates separately) to make these correlations apparent to the reader.

203-204: It is convention to say the dependent variable against the independent variable so “overall acceptance rate plotted against time to first decision”, not the other way around.

234-238: Ahh, I believe I have my answer to 171-172. Perhaps this text is better placed at that location as a caveat (in the Methods).

Fig. 6: Again, advise changing to “Median time to publication versus impact factor”

Fig. 7: I believe I understand what this figure is trying to convey, but it is still unclear as currently formatted. To better discern any trends or long-term changes, I have the following suggestions

• Plot a different colored horizontal line at 1.0 (the baseline)

• Perhaps plot the percentiles (as a ribbon plot?) to clearly summarize how key aspects of the publication time distribution changes over this time frame (e.g. median, inner 50th and 95th)

• Or plot the median, smoother, moving average, or linear fit over these lines and the baseline

6. PLOS authors have the option to publish the peer review history of their article (what does this mean?). If published, this will include your full peer review and any attached files.

Reviewer #1: No

Reviewer #2: No

---

## [Author Response · Author response to Decision Letter 0]

8 Sep 2021

I am grateful to the reviewers and editorial staff for the helpful comments and suggestions. I have replied to each of them in the response letter, and have addressed a few suggestions that I declined to make within the cover letter.

---

## [Editor Report · Decision Letter 1]

13 Sep 2021

Time to publish? Turnaround times, acceptance rates, and impact factors of journals in fisheries science

PONE-D-21-22019R1

Dear Dr. Runde,

We’re pleased to inform you that your manuscript has been judged scientifically suitable for publication and will be formally accepted for publication once it meets all outstanding technical requirements.

Kind regards,

Charles William Martin

Academic Editor

PLOS ONE

Additional Editor Comments (optional):

It is my opinion that your revised manuscript addresses clear and meaningful hypotheses regarding scientist’s decision-making for research product outlets and the relationships for publication selection criteria with important factors such as turnaround time, rates of acceptance, and impact factors. I appreciate the additional text on the drawbacks and limitations to the use of impact factors (the utility of impact factors have often been debated and criticized) in the discussion and think that this improves the manuscript. Regarding PLOS ONE’s publication criteria, all analyses were sound, well-described, and sufficiently detailed and presented in a logical fashion. The data, analyses, and findings of this manuscript will be of use to many in the scientific community (both from the author and publisher standpoint), especially those in the field of fisheries sciences.

---

## [Editor Report · Acceptance letter]

15 Sep 2021

PONE-D-21-22019R1 

Time to publish? Turnaround times, acceptance rates, and impact factors of journals in fisheries science 

Dear Dr. Runde:

I'm pleased to inform you that your manuscript has been deemed suitable for publication in PLOS ONE. Congratulations! Your manuscript is now with our production department. 

Kind regards, 

on behalf of

Dr. Charles William Martin 

Academic Editor

PLOS ONE